# Pathogenic Roles of RNA-Binding Proteins in Sarcomas

**DOI:** 10.3390/cancers14153812

**Published:** 2022-08-05

**Authors:** Yu Hai, Asuka Kawachi, Xiaodong He, Akihide Yoshimi

**Affiliations:** 1Cancer RNA Research Unit, National Cancer Center Research Institute, Tokyo 104-0045, Japan; 2Department of Physical and Chemical Inspection, School of Public Health, Cheeloo College of Medicine, Shandong University, Jinan 250012, China

**Keywords:** RNA-binding protein, sarcoma, splicing, polyadenylation, translation, non-coding RNA, Ewing sarcoma, rhabdomyosarcoma, synovial sarcomas

## Abstract

**Simple Summary:**

DNA information can be copied into mRNA (this process is called transcription) and proteins can be subsequently synthesized using the information in mRNA as a template (called translation). Approximately 4000 RNA-binding proteins (RBPs) in the cells coordinately regulate these multiple processes between transcription and translation. It has been recently recognized that some of the RBPs have abnormal expression and/or function, leading to the initiation or maintenance of malignant disorders including sarcomas, which is the general term for a broad group of malignancies that begin in the bones and soft tissue. Unfortunately, there are currently very few effective treatments for many types of sarcomas in advanced stages. Therefore, we need to understand more deeply how sarcomas develop in our body and how they are efficiently eradicated by therapeutic intervention. Studies on the disease mechanisms in terms of RBPs will provide us with the opportunity to have a better understanding of the sarcoma pathogenesis.

**Abstract:**

RNA-binding proteins (RBPs) are proteins that physically and functionally bind to RNA to regulate the RNA metabolism such as alternative splicing, polyadenylation, transport, maintenance of stability, localization, and translation. There is accumulating evidence that dysregulated RBPs play an essential role in the pathogenesis of malignant tumors including a variety of types of sarcomas. On the other hand, prognosis of patients with sarcoma, especially with sarcoma in advanced stages, is very poor, and almost no effective standard treatment has been established for most of types of sarcomas so far, highlighting the urgent need for identifying novel therapeutic targets based on the deep understanding of pathogenesis. Therefore, defining the network of interactions between RBPs and disease-related RNA targets will contribute to a better understanding of sarcomagenesis and identification of a novel therapeutic target for sarcomas.

## 1. Introduction

RNA-binding proteins (RBPs) are involved in multiple steps of RNA metabolism including RNA splicing, polyadenylation, transport, maintenance of RNA stability and degradation, intracellular localization, and translation control. There is an increasing amount of evidence that alterations in the expression or activity of RBPs can lead to a variety of disorders including cancers and sarcomas. Defining the network of interactions between RBPs and disease-associated RNA targets will contribute to a better understanding of tumor pathology and a development of novel therapeutic agents.

RBPs generally have conserved domains and common sequences that interact with RNA and other proteins to form ribonucleoprotein (RNP) complexes. RBPs determine the fate of target RNAs by recruiting a variety of proteins and other cellular components [1]. RBPs were originally identified for their ability to bind various types of RNA through the formation of stable RNA-binding domains (RBDs) of one or more secondary and tertiary structures, which in turn affect gene expression. Currently, approximately 50 different RBDs have been identified to classify RBPs. Classical RBDs include K homologous domain (KH), RNA recognition motif (RRM), zinc finger domain (ZNF), Pumilio homologous domain (PUM), and double-stranded RNA-binding domain (dsRBD) [2]. Interestingly, only one-quarter of these RBPs contain classical RBDs while the remainder contain nonstandard RBDs with previously uncharacterized motifs [3]. Other unconventional RBPs, which lack recognizable RBDs but typically contain intrinsically disordered regions or mononucleotide- and dinucleotide-binding domains directly involved in RNA binding, are known as “enigma RBPs” because the known cellular biological functions of these proteins are unrelated to RNA biology [4]. The combination of the versatility and structural flexibility of RBDs enables RBPs to control the metabolism of many transcripts. RBPs play a critical role in the regulation of gene expression through post-transcriptional regulation such as alternative splicing, selective polyadenylation, subcellular localization, stability, and RNA translation, as we review later [5,6].

Recent intensive genomic sequencing studies in malignant disorders have revealed recurrent mutations and aberrant activation and deactivation of RBPs due to both genetic and nongenetic factors, suggesting essential roles of RBPs in the initiation and progression of tumors. In addition, advanced technologies assessing RNA–protein interactions have identified more than 4000 proteins as potential RBPs. These advances associated with RBPs and RNA have greatly enhanced basic and translational research on how these RBPs contribute to tumorigenesis.

In fact, it is now clear that RBPs are dysregulated in several types of cancer, thereby affecting the expression and function of oncoproteins and/or tumor suppressor proteins [7,8]. Clinicopathological and immunohistochemical studies of cancer cases have shown that some RBPs are abnormally expressed in cancer relative to the surrounding normal tissues, and such aberrant expression is correlated with the prognosis of patients [9,10,11]. For example, Wang et al. identified RBM39 as a differentially expressed RBPs in acute myeloid leukemia (AML) compared with normal human CD34^+^ hematopoietic stem and progenitor cells [12]. A domain-focused CRISPR/Cas9 screen uncovered a network of interacting RBPs in AML that are essential for RNA splicing and AML survival including RBM39, which can be targeted by sulfonamides (including indisulum (also known as E7070), E7820, and chloroquinoxaline sulfonamide) [12,13].

On the basis of recent advances in the understanding of RBPs in the pathogenesis of malignant tumors, we provide a concise review of RBPs covering their functional roles in RNA metabolism and dysregulation in sarcomas.

## 2. Regulations of RNA Metabolism by RBPs

RBPs are involved in almost all layers of post-transcriptional regulation (RNA metabolism) (Figure 1). RBPs establish highly dynamic interactions with both coding and noncoding RNAsm as well as multiple proteins to achieve these complicated metabolic processes. Although RBPs bind to various types of RNAs including messenger RNA (mRNA), ribosomal RNA (rRNA), small nuclear RNA (snRNA), small nucleolar RNA (snoRNA), and transfer RNA (tRNA), more than one-half of RBPs bind to mRNA to regulate its fate [14]. In this section, we review the regulatory mechanisms of mRNA and noncoding RNA metabolism by several RBPs in the context of normal and malignant cells. 

### 2.1. Alternative Splicing

Alternative splicing (AS) plays an important role in gene regulation, through which pre-mRNA transcripts are processed to produce multiple mRNA variants with different stability and protein-coding potential. AS perturbations are frequently observed across cancers. RBP is one of the molecular determinants of AS, and the disturbance of RBP network activity has a causal relationship with cancer development [15,16]. Functionally aberrant RBPs induce splice isomer conversions, which are widely involved in the regulation of cancer phenotypes including proliferation, apoptosis, cell-cycle progression, invasion and metastasis, angiogenesis, abnormal energy metabolism, and immune escape [17]. The main types of abnormal splicing in tumors are constituent splicing, exon skipping or inclusion, substitution of the 5′ or 3′ splice site, intron retention, and mutually exclusive exon [18]. Classical splicing regulators include serine–arginine (SR) proteins (such as SRSF2) and heterogeneous nuclear ribonucleoproteins (hnRNPs), which bind to exon or intron regulatory elements to promote or prevent the recognition of the 5′ splicing site by the spliceosomal U1 small nuclear ribonucleoprotein (snRNP). These proteins can also promote or block SF1, U2AF2, U2AF1, or U2 snRNP, thereby influencing the selection of splice sites and thus altering splicing decisions [17]. The aberrant expression of SR and hnRNP proteins in cancers suggests that dysregulation of these two types of splicing factors plays an important role in tumor progression. In addition, RNA splicing is frequently dysregulated in a variety of cancers, and hotspot mutations affecting key splicing factors, SF3B1, SRSF2, and U2AF1 are commonly enriched across cancers [19,20], strongly suggesting that aberrant RNA splicing is a new class of hallmark that contributes to the initiation and/or maintenance of cancers. These mutations in genes encoding splicing factors are commonly identified in a variety of hematologic malignancies [19,20,21], as well as in solid tumors such as breast cancers, lung cancers, and pancreatic cancers [22,23,24,25,26]. The pathogenic roles of recurrent mutations affecting these splicing factors [27,28,29,30] and the therapeutic strategies against cancers bearing these mutations [12,27,31] have been extensively studied and reviewed elsewhere [32,33,34].

### 2.2. Alternative Polyadenylation

Polyadenylation is a key process for the generation of mature RNA transcripts. Selective polyadenylation occurs within the 3′ UTR of mRNA and produces 3′ UTR of varying length by 3′-terminal cleavage and polyadenylation (CPA). Each transcript of mature mRNA contains a polyadenylate tail that determines its stability. RBPs involved in polyadenylation include U1 small nuclear ribonucleic protein (U1 snRNP), cleavage and polyadenylation-specific factor 1 (CPSF1), embryonic lethal abnormal vision (ELAV) L1/human antigen R (HuR), and poly(A) RNA-binding protein (examples include the cytoplasmic polyadenylate element-binding protein family, CPEB1–4), and the ZFP36 ring finger protein (ZFP36/TTP) [35]. CPEB family proteins recruit the translational inhibitory factors or cytoplasmic polyadenylation factors and regulate the length of the poly(A) tail. The C-terminal region of CPEB family proteins contains two RRM and two zinc finger-like motifs, as well as a variable N-terminal region [36]. For example, TP53 mRNA contains CPE in its 3′-UTR, which promotes polyadenylation. Burns et al. showed that CPEB1 enhanced TP53 mRNA polyadenylation and translation with the cytoplasmic poly(A) polymerase GLD-4 [37]. The authors demonstrated that TP53 mRNA has a short poly(A) tail and a reduced translational efficiency, leading to a decrease in p53 protein expression [38]. 

### 2.3. Stability

RNA stability is associated with its nucleotide sequence, modification, 5′ m^7^G cap, and 3′ poly (A) tail [39]. These determinants for RNA stability regulate the mRNA decay and translation initiation [40]. More specifically, mRNA degradation is mainly regulated by the following two mechanisms: (1) one mechanism starts with deadenylation of the 3′ poly (A) tail, which is followed by 5′ cap removal and 5′-to-3′ decay; (2) the other mechanism begins after hydrolysis of the 3′ poly (A) tail and 3′-to-5′ decay [40].

Several RBPs such as the mRNA decapping enzyme scavenger (DCPS), CUGBP Elav-like family member 2 (CELF2), insulin-like growth factor 2 mRNA protein (IGF2BP) family, HuR, QKI-5, RBMS3, and TARBP2 play important roles in cancer biology [6]. For example, Yamauchi et al. performed a genome-wide CRISPR/Cas9 screen in murine AML models with the oncogenic fusions CALM/AF10 and MLL/AF9 and identified DCPS as a promising target for AML [41,42]. The 5′ end of eukaryotic mRNA is characterized by a distinctive “cap”, which consists of an N7 methylated guanine (m^7^GpppN). The 5′ cap is important for promoting splicing of the first intron, exporting mRNA to the cytoplasm, and allowing translation of the mRNA. Removal of the cap (“decapping”) results in silencing of mRNA expression. The decapping enzyme DCPS is characterized as a pyrophosphatase that hydrolyzes the 5′ m^7^Gppp and m^7^Gpp cap structure generated following 3′-to-5′ and 5′-to-3′ decay. Messenger RNAs containing an AU-rich element (ARE) in the 3′ UTR are rapidly degraded in the cytoplasm. ARE-mediated decay is initiated by deadenylation, which is followed by the 3′-to-5′ decay through a complex of exonucleases (termed exosome). In this process, DCPS hydrolyzes the remaining cap [43]. On the other hand, the 5′-to-3′ decay pathway is initiated by cleavage of the 5′ cap structure to release the m^7^Gpp and monophosphorylated RNA, followed by the DCPS-mediated hydrolysis of m^7^Gpp to release m^7^Gp [44]. Therefore, DCPS plays an essential role in the final step of removing the residual cap in both directions of RNA decay. Intriguingly, DCPS was shown to be dispensable for normal hematopoiesis, which was supported by the observation that clinical parameters including blood cell counts in persons with germline biallelic loss-of-function mutations in DCPS were not significantly affected. In summary, DCPS is potentially an AML-selective vulnerability for which development of a targeted therapy is expected.

### 2.4. RNA Localization

Another layer of regulation on the stability, as well as the translation, of mRNA is achieved by RBP-mediated control on intracellular localization [45]. Subcellular localization of mRNAs involves several steps and requires the coordinated involvement of multiple protein factors. Cis-motifs and postcode elements in mRNA 3′ UTR are the important factors for RBPs to coordinately involve this regulatory process [46]. One of the best examples is IGF2BP1 (also known as IMP1/ZBP1), which is a member of the conserved VICKZ family of RBP. IGF2BP1 controls cell adherence and polarity in breast cancer by physically binding to a subset of mRNAs that encode important mediators such as E-cadherin, β-actin, α-actinin, and ARP-16 for these cellular properties [47].

### 2.5. Translation

The translation process consists of the following three steps: initiation, extension, and termination. Controls on these processes are crucial for cancer development and progression both globally and in specific mRNAs which promote cancer biology, such as cell survival, transformation, metastasis, and stemness. Multiple factors including 43S ribosome initiation complex, cap-dependent mRNA translation, and cap-independent mRNA translation are involved in carcinogenesis. Dysregulations in these translational processes in malignant disorders are beyond the scope of this review and are excellently summarized elsewhere [48,49,50].

### 2.6. Noncoding RNA Processing

No-coding RNAs (ncRNAs) are commonly expressed RNAs in human cells that lack protein-coding ability. In the research results of FANTOM and ENCODE, two large genome projects, 80% of the human genome has transcriptional activity, while only 2% of the human genome codes for proteins. The noncoding regions of the human genome (98%) are primarily responsible for regulating gene expression [51,52]. RBPs also bind to noncoding RNAs such as microRNAs (miRNA), transfer RNAs (tRNA), siRNA, telomerase RNA, small nucleolar RNA (snoRNA), and splicing small nucleolar RNA (snRNA), which regulates multiple molecular processes including RNA splicing and modification, protein localization, and chromosomal remodeling [53]. The interactions between noncoding RNAs and RBPs are increasingly recognized as one of the basic mechanisms of gene regulation and plays a crucial role in cancer [54]. RBPs are key regulators of miRNA biogenesis and maturation. They promote or inhibit miRNA processing mainly through their effects on typical proteins such as Drosha and Dicer. Altered functions in such RBPs result in the impairment of miRNA processing, which in turn affects expression of cancer-associated genes [55].

The RBP LIN28 is one of the four factors sufficient to reprogram human somatic cells into induced pluripotent stem cells, which upregulates or inhibits the maturation of different members of the let-7 microRNA family in many cancer cells [56]. LIN28A binds to the terminal ring of precursor let-7 (pre-let-7) and enforces terminal uridine transferase (TUTase) ZCCHC11, which can acidify pre-let-7 polyuridine, thereby blocking miRNA biogenesis and tumor suppressor function. For LIN28B, the exact mechanism that causes let-7 inhibition remains controversial. Functionally, the reduction in let-7 miRNA leads to the overexpression of its oncogenic targets such as MYC, RAS, HMGA2, and BLIMP1 [57]. In addition to let-7, other miRNAs (miR-9, -107, -143, -200C, -370, and -638) containing the same tetranucleotide sequence motif (GGAG) as pre-let-7 undergo the same process. Most of these miRNAs have been identified as tumor suppressors, suggesting that LIN28 may promote cancer metastasis by inhibiting multiple metastasis-associated miRNAs [6].

CircRNAs are covalently closed RNA molecules, usually derived from precursor mRNA (pre-mRNA) through reverse splicing events, where the downstream 5′ splicing donor is reversely spliced to the upstream 3′ splicing receptor [58]. Recent evidence suggests that abnormal circRNA expression exists in almost all cancer types and plays an indispensable role in cancer pathogenesis as oncogenes or tumor suppressors [59]. CircRNA can act as a protein scaffold or antagonist that interacts with RBPs. For example, circACC1 acts as a protein scaffold to enhance the enzyme activity of the AMP-activated protein kinase (AMPK) holoenzyme by directly binding AMPK β and γ subunits. [60]. CircRHOBTB3 also inhibits metastasis in colorectal cancer by interacting with HuR, which in turn degrades HuR to reduce the expression level of the downstream target PTBP1 [61].

## 3. Role of RBPs in Sarcomas

In this section, we review some representative types of sarcomas in which RBPs play essential pathogenic roles. Importantly, some of these are being explored as therapeutic targets. We comprehensively review the targetable genes by sarcoma types from The Cancer Dependency Map Project at Broad Institute (DepMap: https://depmap.org/portal/ (accessed on 1 June 2022)) and list the top-ranked RBPs (Table 1).

### 3.1. Ewing Sarcoma

Ewing sarcoma is a highly aggressive type of sarcoma, mostly affecting children. The 5 year overall survival is 70% in localized cases with intensive multidisciplinary treatment and 15–30% in advanced cases [62]. Standard therapy for advanced cases has not yet been established. Ewing sarcoma is molecularly characterized by an oncogenic chimeric aberrant transcription factor arising from a chromosome translocation, which fuses the transactivation (TAD) domain of EWS RNA-binding protein 1 (EWSR1) to the DNA-binding domain of ETS family genes [63]. Among the diverse ETS family genes, FLI1 is the most common fusion partner, accounting for 80–90% cases of Ewing sarcoma [63]. Given the fact that only 20% of Ewing sarcoma cases are accompanied by additional gene mutation, the aberrant transcription factor may have the ability to solely drive tumor phenotype [64]. 

EWS is a member of the TET family of RBPs and participates in the diverse aspects of RNA metabolism to regulate DNA repair, cell proliferation, senescence, etc. [65]. In addition to these functions on RNA processing, the role of TAD in EWS containing low-complexity sequence domains (LADs) has recently been shown to facilitate liquid–liquid phase transition [66]. Chong et al. discovered that the presence of TAD is crucial for segregating EWS/FLI1 on its binding sites on the DNA sequence. The optimum concentrations of EWS/FLI1 and wildtype EWS in liquid–liquid phase transition are regulated within a narrow range to enhance the oncogenic transcription programs driven by EWS/FLI1. In addition, the TAD of EWS possesses prion-like domains, through which EWS interacts with the SWI/SNF complex, as well as RNA polymerase II. EWS/FLI1 establishes super-enhancers by recruiting the SWI/SNF complex to the target genes and dramatically increases transcriptional activity [67,68,69,70]. To summarize, the phase separation model may explain the mechanisms of super-enhancer formation mediated by combined EWS/FLI1, SWI/SNF complex and RNA polymerase II. 

Recently, Julien et al. discovered that some genes are uniquely transcribed by chimeric transcription factors, and their distinct transcription, mRNA processing, and translational pattern generates tumor neoantigens in oncogenic chimeric protein-driven sarcomas including Ewing sarcoma [71]. This fusion gene-mediated dysregulation on these molecular consequences leads to survival advantage of sarcomas. For example, ARID1A, a subunit of the SWI/SNF complex, is transcribed into several isoforms. EWS/FLI1 preferentially increases the ARID1A-L isoform over the ARID1A-S isoform. EWS/FLI1 directly interacts with the SWI/SNF complex through ARID1A-L and regulates target gene expression, resulting in promoting tumor growth (Figure 2) [72]. 

Some other well-known RBPs also contribute to the Ewing sarcoma pathogenesis. The LIN28 paralog is canonically known to regulate gene expression through let-7 miRNA family biogenesis. Keskin et al. discovered that LIN28B is highly expressed in a subset of Ewing sarcoma [73]. LIN28B enhanced tumor growth by stabilizing the mRNA of EWS/FLI1 independently of let-7 (Figure 2). According to the results from the DepMap project, the IGF2BP family is a potential therapeutic target of Ewing sarcoma (Table 1). Knockout/knockdown of IGF2BP significantly inhibited tumor growth in vivo. Clinically, patients with high IGF2BP expression showed an unfavorable prognosis compared to patients with low IGF2BP expression [74]. IGF2BP is transcribed to three major isoforms, IGF2BP-1, -2, and -3. IGF2BP is predominantly localized in the cytoplasmic fraction and is considered to stabilize pre-mRNA as a reader of N^6^-methyladenosine (m6A), inhibit RNA decay, and enhance the expression of oncogenes such as MYC and LIN28 in multiple types of cancers [75]. Thus, IGF2BP inhibitors have been intensively investigated.

### 3.2. Rhabdomyosarcoma

Rhabdomyosarcoma (RMS) is the most common soft-tissue sarcoma in children and rarely affects adults. Pathologically, RMS is a high-grade mesenchymal tumor showing aberrant myogenic differentiation. The 5 year survival rates are approximately 60% in children and 30% in adults [76]. A multidisciplinary therapy combining local surgery and cytotoxic agents was established in 2009 as the standard of care [77]. Molecular targeted therapy is under investigation based on novel molecular findings. The embryonal and alveolar types of RMS are the two major pathologically distinct subtypes. Alveolar RMS usually contains a balanced chromosomal translocation of t(2;13)(q35;q14) or a variant t(1;13)(p36;q14), both of which generate oncogenic fusion proteins. On the other hand, such an oncogenic fusion is absent in embryonal RMS. Embryonal RMS and fusion-negative alveolar RMS have overlapped expression profiles, while fusion-positive alveolar RMS harbors a distinct molecular profile [76].

Targeting the insulin growth factor (IGF) pathway is considered an attractive therapeutic target for RMS [78,79]. RMS has a survival advantage by switching isoforms of IR, the receptor for insulin growth factor. IR-A, which lacks exon 11 of IR-B, has a greater affinity for the ligand than IR-B and is predominantly expressed in RMS tissue [80]. The mechanism of this isoform switching remains unknown.

According to the DepMap database, Quaking (QKI) was identified as a candidate of the therapeutic target of RMS (Table 1). Although the mechanism is unknown, inhibition of QKI expression greatly suppressed the growth of RMS cell lines. QKI is an RBP which belongs to the signal transduction and activation RNA (STAR) family [61]. There are three major alternatively spliced isoforms, QKI-5, QKI-6, and QKI-7, which have differential carboxy-terminal domains. Although these isoforms share a common RNA-binding property, each isoform uniquely regulates pre-mRNA splicing, transportation, and stability in a cell type-specific manner. QKI-5, the most abundant splicing isoform of the human QKI gene, has been identified as a functional target of miR-221. Yang et al. identified 2014 variable triplets, including 1101 modulators and 187 splicing events. Pathway enrichment analysis showed that QKI splicing targets were enriched in tightly connected pathways, i.e., the endocytosis and MAPK signaling pathways, on which a specific proportion of RMS cells are dependent [81].

### 3.3. Synovial Sarcoma

Synovial sarcomas account for 8–10% of all soft-tissue sarcoma cases and most commonly affect the AYA generation younger than 30 years old. Clinical prognosis of adult synovial sarcoma remains poor; the expected 5 year survival is 50–60% despite intensive multimodality treatment [82]. Synovial sarcoma is molecularly characterized by translocation t(X;18)(p11;q11), which forms a fusion gene between SS18 (SYT) on chromosome 18 and SSX1, SSX2, or SSX4 on chromosome X. SS18 is a subunit of SWI/SNF (BAF) complexes, and the SS18/SSX fusion protein alters SWI/SNF composition, in which a tumor suppressor, SMARCB1, is lacking [83]. The SWI/SNF complex is a chromatin remodeler and regulates gene expression together with polycomb repressive complexes. The aberrant SWI/SNF complex dysregulates immune evasion, cellular plasticity, and cell cycle in synovial sarcoma [84]. Recent research showed that SMARCB1 interacts with lncRNA and controls target gene expression at active promoter regions. An RNA immunoprecipitation assay identified that SMARCB1 is bound to both coding RNA and long noncoding RNA (lncRNA) at active promoter regions. For example, SWINGN, a lncRNA, modulates a large network of pro-oncogenic genes by favoring SMRCB1 binding [85]. Although the alteration of RNA interaction with aberrant SWI/SNF in synovial sarcoma remains greatly unknown, disruption of the interaction may contribute to = oncogenic transcription.

### 3.4. Osteosarcoma

Osteosarcoma is the most common bone-derived malignant tumor, affecting mostly adolescents. The clinical prognosis is dramatically improved by multimodality therapy; long-term survival is obtained in >60% of patients with localized disease. However, limited progress has been achieved over the last four decades. Histologically, osteosarcoma is characterized by an abundant osteomatrix synthesized by malignant cells and classified as osteoblastic, chondroblast, or fibroblastic according to the predominant differentiation pattern [86]. Recent intensive genomic sequencing revealed that the oncogenic events of 75% of osteosarcoma cases was explained by chromothripsis, which constitutes massive, clustered genomic rearrangements. Recurrent somatic gene mutations are frequently detected in *TP53* (95%), *RB* (29%), *ATRX* (52%), and *DLG2* (52%) [87].

The role of RBPs in the osteosarcoma phenotype is poorly understood. A few RBPs have been reported as potentially relevant to clinical prognosis and cell proliferation. SUB1/PUC4, a coactivator of RNA polymerase II, was identified as a potential targetable vulnerability of osteosarcoma in the DepMap database (Table 1). The molecular roles of SUB/PUC4 are associated with transcription, DNA repair mand replication. Hu et al. reported that patients with high SUB1/PUC4 expression showed unfavorable prognosis compared to patients with low expression. Inhibiting PUC4 expression reduced the potential for lung metastasis [88].

## 4. Conclusions

Although our understanding of the genomics, molecular biology, and therapeutic implications of altered RBPs in a variety of cancers and sarcomas has been greatly improved by the recent elegant studies using cutting-edge techniques such as single-cell RNA/DNA sequence and CRISPR/Cas9 mediated screens, standard treatment strategies targeting sarcomas in advanced stages have been only partly established, which underscores an urgent need for the development of novel therapeutics for sarcomas. In addition, the full contribution of aberrant RBPs to sarcomagenesis has not yet been elucidated. Interestingly, driver fusion genes in sarcomas favor RBPs with LCDs such as EWS/FLI1 and TAF15/NR4A3 [89]. Although the molecular reasons that sarcomas prefer the usage of RBPs as their drivers remain unknown, EWS/FLI1, for example, recruits the SWI/SNF complex to directly establish super-enhancers on the loci different from the binding sites of wildtype EWS (this process is mainly achieved by the TAD of EWS), which is also supported by LAD-mediated phase separation. In addition, EWS/FLI1 functions as a splicing modulator to preferentially enhance the ARID1A-L isoform expression, which strengthens the physical binding between EWS/FLI1 and SWI/SNF. Such multiple functions of RBPs in combination with the molecular background of origin of cells/tissues may be one of the reasons for the marked dependency of sarcomas on RBP.

There remain several unsolved problems in terms of the pathogenesis of sarcomas and the development of therapeutic avenues for sarcomas with functionally aberrant RBPs. On the other hand, the identification of RBM39 as a dependency molecule in AML, for example, is a promising result as RBM39 is directly targeted by sulfonamides, providing a unique example of translation from a basic RBP screen to the development of pharmacological intervention against malignancies.

In summary, this review highlights the functional and pathological roles of RBPs in the initiation and maintenance of sarcomas and other malignant disorders. We are just beginning to understand how aberrant RBPs contribute to tumorigenesis and how we could therapeutically target aberrant RBPs in tumors, indicating the need for further studies.

## Figures and Tables

**Figure 1 cancers-14-03812-f001:**
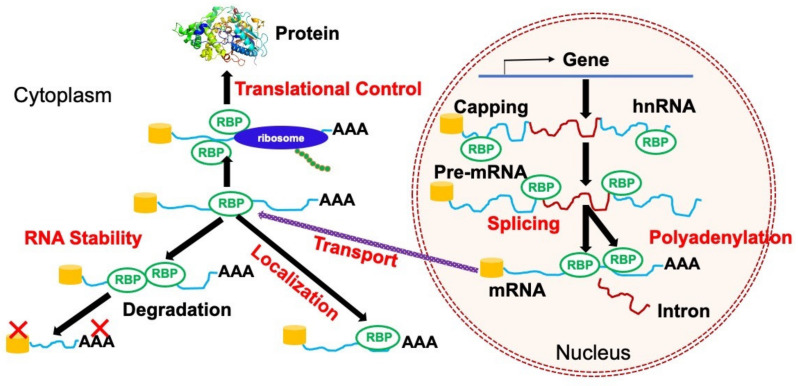
A scheme showing the roles of RNA-binding proteins (RBPs) in multiple layers of post-transcriptional regulations. RBPs are involved in a variety of RNA metabolic processes including splicing, polyadenylation, transport, localization, RNA stability, and translational control.

**Figure 2 cancers-14-03812-f002:**
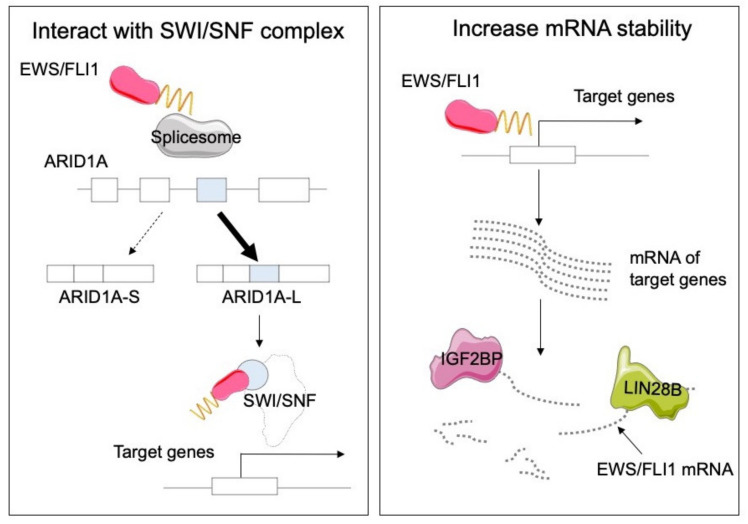
Interaction and stabilization by the EWS/FLI1 fusion protein. EWS/FLI, together with the SWI/SNF complex, upregulates the expression of Ewing sarcoma-specific target genes via multiple mechanisms. An alternative splicing event in ARID1A in cells with EWS/FLI1 fusion enhances direct binding between SWI/SNF complex and EWS/FLI1, which promotes the growth of Ewing sarcoma cells. On the other hand, RBPs including LIN28B and IGF2BP are activated in EWS/FLI1-positive cells and may play essential roles in stabilizing mRNA.

**Table 1 cancers-14-03812-t001:** Top-ranked RNA-binding proteins (RBPs) in sarcomas according to the DepMap project.

Sarcoma	RNA Binding Proteins	T-Statistic	*p*-Value
Ewing sarcoma	EWSR1	−13.16812526	1.03 × 10^−36^
	IGF2BP1	−12.2904187	1.70 × 10^−32^
	STAG1	−11.52248821	5.57 × 10^−29^
Rhabdomyosarcoma	QKI	−8.65279311	2.29 × 10^−17^
Synovial sarcoma	SRSF2	−5.668685512	2.09 × 10^−8^
	LSM14B	−4.679259416	3.26 × 10^−6^
Osteosarcoma	SUB1	−5.501544322	4.92 × 10^−8^
	ASCC3	−4.133181793	3.91 × 10^−5^
Liposarcoma	DAZAP1	−5.33256277	1.19 × 10^−^^7^

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
