# Peer review of "Pathogenic Roles of RNA-Binding Proteins in Sarcomas"

_cancers, 2022, doi:10.3390/cancers14153812_

Round 1
Reviewer 1 Report
The Reviewer’s comments: The manuscript entitled “Pathogenic Roles of RNA Binding Proteins in Sarcomas” contributed by Yu Hai, Asuka Kawachi, Xiao-Dong He, and Akihide Yoshimi provides an informative, exiting and comprehensive piece of infomation over recent research activities of the roles of RBPs in generating sarcomas. This will give a beneficial and supportive review article of the sarcoma research and also stimulate readers’ interests in it, especially young scientists. The reviewer would say it should be a great boost for the field of the sarcoma study. The description is well organized and rigorously selected for a concise space of the publication.
Only thing the reviewer would ask the authors to give us an idea of any speculation why sarcoma employs RBPs to generate and develop themselves in the human body. One clue is to generate the fusion genes with specific transcription factors. Another topic might be the fusion gene product bearing intrinsically disordered region in them. The reviewer would like to have this sort of speculative description near the end of the manuscript. This will assist active researchers with interests in the sarcoma biology.
Author Response
The Reviewer’s comments: The manuscript entitled “Pathogenic Roles of RNA Binding Proteins in Sarcomas” contributed by Yu Hai, Asuka Kawachi, Xiao-Dong He, and Akihide Yoshimi provides an informative, exiting and comprehensive piece of infomation over recent research activities of the roles of RBPs in generating sarcomas. This will give a beneficial and supportive review article of the sarcoma research and also stimulate readers’ interests in it, especially young scientists. The reviewer would say it should be a great boost for the field of the sarcoma study. The description is well organized and rigorously selected for a concise space of the publication.
REPLY: We thank the Reviewer for the kind compliments that the manuscript “provides an informative, exiting and comprehensive piece of infomation over recent research activities of the roles of RBPs in generating sarcomas” and “should be a great boost for the field of the sarcoma study.”
Only thing the reviewer would ask the authors to give us an idea of any speculation why sarcoma employs RBPs to generate and develop themselves in the human body. One clue is to generate the fusion genes with specific transcription factors. Another topic might be the fusion gene product bearing intrinsically disordered region in them. The reviewer would like to have this sort of speculative description near the end of the manuscript. This will assist active researchers with interests in the sarcoma biology.
REPLY: We thank the Reviewer for this question and have now described our speculation on the reasons that sarcoma specifically employs RBPs, especially focusing on the example of EWSR1-FLI1 fusion as follows:
In line 247
In addition to these functions on RNA processing, the role of TAD in EWS containing low-complexity sequence domains (LADs) has been recently shown to facilitate liquid-liquid phase transition [68]. Chong et al. discovered that the presence of TAD is crucial for segregating EWS/FLI1 on its binding sites on DNA sequence. The optimum concentrations of EWS/FLI1 and wild-type EWS in liquid-liquid phase transition are regulated within a narrow range to enhance the oncogenic transcription programs driven by EWS/FLI1. In addition, the TAD of EWS possesses prion-like domains, through which EWS interacts with SWI/SNF complex as well as RNA polymerase II. EWS/FLI1 establishes super-enhancers by recruiting SWI/SNF complex to the target genes and dramatically increases transcriptional activity [69-72]. To summarize, the phase separation model may explain the mechanisms of super-enhancer formation mediated by combined EWS/FLI1, SWI/SNF complex and RNA polymerase II.
In line 364 (in the Conclusions session)
Interestingly, driver fusion genes in sarcomas favor RBPs with LCDs such as EWS/FLI1 and TAF15/NR4A3 [91]. Although the molecular reasons that sarcomas prefer the usage of RBPs as their drivers remain unknown, EWS/FLI1 for example recruits SWI/SNF complex to directly establishes super-enhancers on the loci different from the binding sites of wild-type EWS (this process is mainly achieved by the TAD of EWS), which is also supported by the LAD-mediated phase separation. In addition, EWS/FLI1 functions as a splicing modulator to preferentially enhance the ARID1A-L isoform expression, which strengthens the physical binding between EWS/FLI1 and SWI/SNF. Such multiple functions of RBPs in combination with the molecular background of origin of cells/tissues may be one of the reasons for the marked dependency of sarcomas on RBP.
Reviewer 2 Report
RNA and RNA binding proteins are gaining an increasing interest for their role in the regulation of gene expression. Summarising the accumulated knowledge on the involvement of RBPs in different malignancies, like specific cancers is important and has the capacity to inspire further research. Therefore the topic of this manuscript is relevant and timely.
The manuscript is logically organised and generally well-written. The role of RBPs in general and in different sarcomas are introduced separately which makes the digestion of the information easy. The figures complement the text nicely and are good quality.
However, several minor grammatical errors can be found in the text, for example:
Line 93: RBPs are involved
Line 162: degraded
Line 176: control
Line 183: encode
Line 208: RBPs are key regulators
Line 231: RBPs
Line 257: sarcomas
Line 260: This
Line 261: leads
Line 276: stabilize
Figure 2: titles of the panels: Interaction and Stabilisation
Line 283: of Ewing's sarcoma-specific target genes
Line 320: account
Line 331: assay
Line 333: regions
There might be other similar mistakes, so I suggest a thorough grammar and spelling check of the manuscript.
Author Response
RNA and RNA binding proteins are gaining an increasing interest for their role in the regulation of gene expression. Summarising the accumulated knowledge on the involvement of RBPs in different malignancies, like specific cancers is important and has the capacity to inspire further research. Therefore the topic of this manuscript is relevant and timely.
The manuscript is logically organised and generally well-written. The role of RBPs in general and in different sarcomas are introduced separately which makes the digestion of the information easy. The figures complement the text nicely and are good quality.
REPLY: We thank the Reviewer for the kind compliments that “the topic of this manuscript is relevant and timely” and that ”the manuscript is logically organised and generally well-written”.
However, several minor grammatical errors can be found in the text, for example:
Line 93: RBPs are involved
Line 162: degraded
Line 176: control
Line 183: encode
Line 208: RBPs are key regulators
Line 231: RBPs
Line 257: sarcomas
Line 260: This
Line 261: leads
Line 276: stabilize
Figure 2: titles of the panels: Interaction and Stabilisation
Line 283: of Ewing's sarcoma-specific target genes
Line 320: account
Line 331: assay
Line 333: regions
There might be other similar mistakes, so I suggest a thorough grammar and spelling check of the manuscript.
REPLY: We appreciate the careful review from the Reviewer and apologize to the Reviewer that there are several typos. We have edited all the points raised by the Reviewer and others by checking the entire manuscript again.